# Functional impact of cardiac implanted devices on ipsilateral shoulder range of motion, scapular mobility, and self-reported quality of life

Cansu Cosgun[1‡*], Muharrem Said Cosgun[2‡], Oznur Buyukturan[3], Buket Buyukturan[3]

**1** Department of Physical Therapy and Rehabilitation, Mengucek Gazi Education and Research Hospital, Erzincan, Turkey, **2** Medical Faculty, Department of Cardiology, Mengucek Gazi Education and Research Hospital, Erzincan Binali Yildirim University, Erzincan, Turkey, **3** Department of Physical Therapy and Rehabilitation, Kırşehir Ahi Evran University, Kırşehir, Turkey

‡ These authors contributed equally to this work. CC and MSC are Joint Senior Authors
* cansu_karaman@yahoo.com

**Data Availability Statement:** All relevant files are available from the zenodo database (URL: zenodo.org/record/7739107#.ZBJfT3ZBzIU, DOI: 10.5281/zenodo.7739107).

## Abstract

### Purpose

Shoulder disorders may occur for procedural reasons in patients fitted with a cardiac implantable electronic device (CIED). This study aimed to examine the effects of CIED implantation on shoulder functions and scapular dyskinesis.

### Materials and methods

Thirty patients fitted with a CIED formed the study group (SG), whilst 30 participants without a CIED formed the control group (CG). The range of motion (ROM), grip strength, lateral scapular slide test (static), scapular dyskinesis test (dynamic), American Shoulder and Elbow Surgeons (ASES) Shoulder Score, and the Short Form-36 Health Survey (Physical and Mental Component Summary [PCS and MCS]) were applied in the study.

### Results

The shoulder's mean flexion and abduction ROM on the implant side were found to be significantly lower in the SG than the CG ($p = .016$ and $p = .001$, respectively). Similarly, a significant grip strength loss on the implant side was detected in the SG than in the CG ($p = .036$). Static and dynamic scapular dyskinesis frequencies were shown to be significantly higher in the SG than in the CG ($p = .002$ and $p < .001$, respectively). The ASES Shoulder Score and PCS score were significantly lower in the SG than in the CG ($p = .014$ and $p = .007$, respectively). However, no difference was revealed between the two groups with respect to the contralateral upper limb.

**Funding:** The author(s) received no specific funding for this work.

**Competing interests:** The authors have declared that no competing interests exist.

**Abbreviations:** ADL, activities of daily living; ASES, American Shoulder and Elbow Surgeons; BMI, body mass index; CG, control group; CIED, cardiac implantable electronic device; cm, centimeter; kg-f, kilogram-force; LVEF, left ventricular ejection fraction; LSST, lateral scapular slide test; MCS, mental component summary; PCS, physical component summary; QoL, quality of life; ROM, range of motion; SDT, scapular dyskinesis test; SF-36, 36-Item Short-Form Survey; SG, study group.

## Conclusion

The frequency of scapular dyskinesis and disability was higher, and upper limb functions, grip strength, and physical subdivision of quality of life decreased in CIED recipients. These findings suggest that such parameters should be included in physiotherapy assessment and treatment programs.

## Introduction

The number of cardiac implantable electronic device (CIED) implantations increases daily as the population ages, as they are used in the primary treatment and secondary prevention of heart diseases [1–3]. The positive effects of CIED implantation on the cardiovascular system are undeniable [4–6]. Still, although considered a minimally invasive procedure [7], recipients may be adversely affected due to early (acute) and late (chronic) procedural complications [8]. Acute procedural complications such as pneumothorax can be life-threatening, and the literature has sufficiently clarified the avoidance and management of these complications [9,10]. Chronic complications such as shoulder impairment can lead to decreased quality of life (QoL) due to the level of disability that they cause [11,12]. However, these complications that affect the shoulder joint and cause permanent disability have not received sufficient attention in the literature.

Existing studies have shown that early mobilization following CIED implantation is safe [13–16]. However, in daily practice, operators still prefer to restrict ipsilateral upper extremity movements above head level, and to even immobilize the arm on the implant side with a sling for a period of 2 to 3 weeks in order to mitigate the risks of early complications such as post-procedure hematoma and lead dislocation [17,18]. This behavior raises concern for device malfunction in patients who have been recently implanted with a CIED. Patients may extend this period due to device failure anxiety and the antalgic effect of immobilization [19]. In addition, the sensation of tightness and shoulder elevation created by the CIED due to its proximity to the pectoral muscles also causes posture disorders resulting in arm movement limitations [19,20]. The restricted range of motion (ROM) and muscle weakness resulting from this unplanned, prolonged immobilization can negatively affect shoulder functions. Finally, frozen shoulder syndrome and adhesive capsulitis may develop following surgery in close proximity to the shoulder, such as CIED implantation [14,21]. The incidence of this possible but often overlooked complication, seen in approximately half of all CIED recipients [20], gradually decreases up to a year following the operation but can also become permanent and thereby lead to a decline in QoL [12].

The adverse effects of CIED implantation on the ipsilateral glenohumeral joint have been explained in the literature [11,19]. However, it is unclear whether or not the scapulothoracic joint, a functional joint that works in coordination with the glenohumeral joint to maintain optimal scapulohumeral rhythm [22], will be affected by ipsilateral CIED implantation. Although scapular dyskinesis is a visible change in scapulothoracic joint kinematics relative to the thorax due to muscle weakness and instability, it is often overlooked. During the shoulder examination, the scapula examination should not be neglected. When scapular dyskinesis is detected, information that helps in determining the management of the patient's clinic, treatment options, rehabilitation protocols that can be applied, and returning to social life is provided [23]. The current study was designed to address this gap in the literature. This study therefore investigated the shoulder functions, grip strength, scapular dyskinesis, shoulder

functionality—as assessed by pain and activities of daily living (ADL)—on the implant side and QoL in patients with CIED.

## Materials and methods

### Participants and data collection

This single-center, cross-sectional study evaluated patients who received a CIED implantation at least one year previous. The study group data were collected from patients who applied to an outpatient clinic for device control during September and October of 2021. A total of 42 CIED recipients were screened for eligibility for this purpose. Patients with neuromuscular (that can cause immobility, plegia, or paresis), rheumatological (tendinitis or arthritis), or orthopedic (fractures, infections, or tumors) disorders affecting the upper limbs and conditions such as injury or surgery (including mastectomy and hemodialysis fistula) were excluded from the study. Patients undergoing nonroutine device implantations (subpectoral or axillary insertion), generator replacement (for elective replacement or end-of-life), revision (for lead dislocation, hematoma, or infection), or those who had experienced any complications (e.g., pneumothorax or hemothorax) were also excluded from the study. Finally, participants who were pregnant or breastfeeding, or who had an uncontrolled psychiatric or endocrinological disorder were also excluded. After these exclusion criteria had been applied, a total of 30 patients formed the study group (SG), while the control group (CG) consisted of 30 participants with similar demographic and clinical characteristics, but with no history of CIED implantation. The patient registration flowchart is illustrated in Fig 1.

The participants' demographic characteristics (age, gender, dominant side, and literacy status) as well as their clinical data were recorded. During the clinical evaluation, a detailed anamnesis (smoking habits, alcohol consumption, and previous diseases), physical examination (body mass index [BMI]), and transthoracic echocardiography (left ventricular ejection fraction [LVEF]) were performed. Relevant information was retrieved from the CIED implantation operation notes of the SG patients, and a plain chest radiograph was taken so as to determine the implanted device's location.

Our study was designed in line with the ethical guidelines of the Declaration of Helsinki for human subjects. Ethics committee approval was received from the Erzincan Binali Yıldırım University Clinical Research Ethics Committee (decision number 10/21, dated September 29, 2021). Signed informed written consent was obtained from each participant prior to their inclusion in the study.

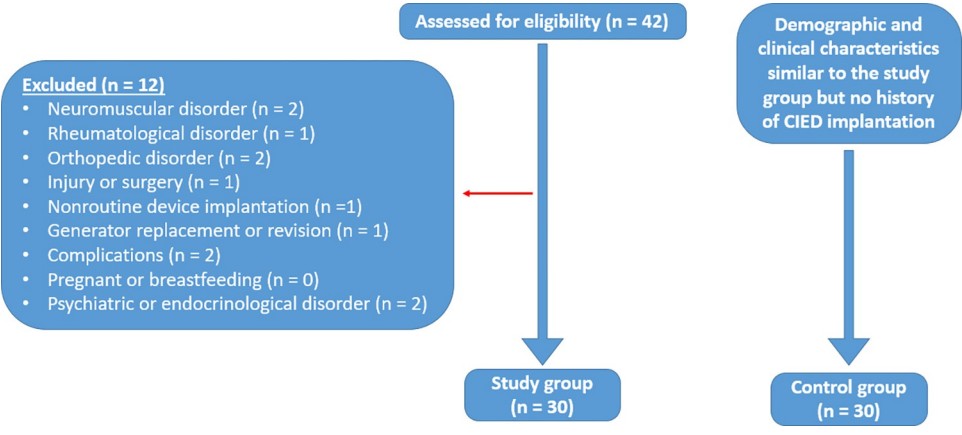

**Fig 1. Study flowchart.**

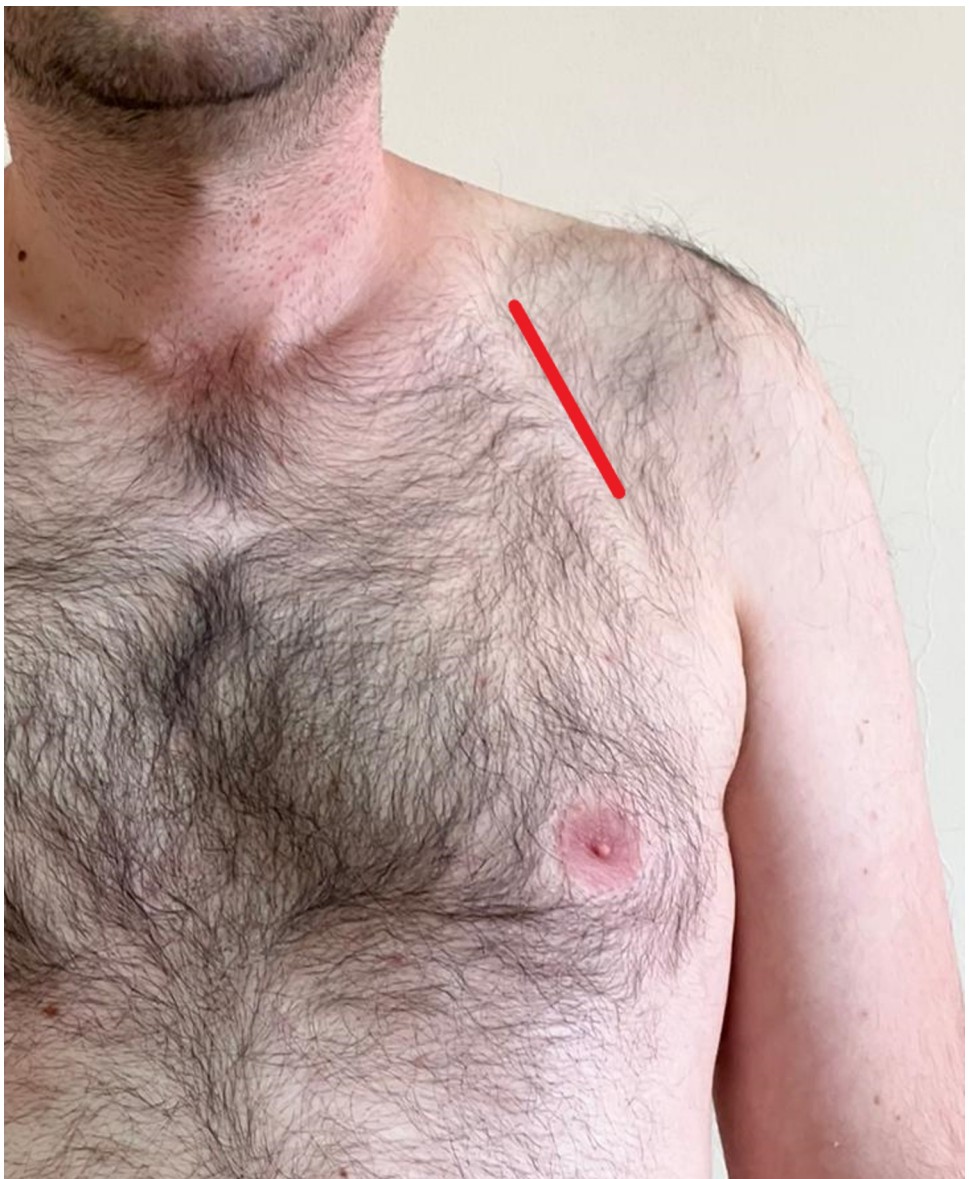

**Fig 2. Demonstration of D-type (deltopectoral) skin incision (red line).**

### Device implantation technique

All of the SG participants had their CIEDs inserted in the left pre-pectoral region by creating a subcutaneous pocket through a 4–5-centimeter (cm) D-type (deltopectoral) skin incision (see Fig 2). The leads were placed transvenously (axillary or subclavian). The subcutaneous tissue and skin were then closed based on the anatomical layers. The patients were each warned against excessive movement and hard work of the ipsilateral shoulder in order to prevent hematoma or lead dislocation.

### Shoulder evaluations

An experienced physiotherapist performed a detailed shoulder assessment of each participant. Shoulder functions were evaluated according to the joint's ROM. The participants' active

shoulder ROM was measured for flexion and abduction using a goniometer without trunk compensation and recorded in degrees (°). All measurements were performed twice, and the average value taken. The Jamar Hydraulic Hand Dynamometer (Sammons Preston, Warrenville, IL, USA) was used to measure the participants' grip strength. Each participant performed 5-second maximal contractions in the sitting position, with shoulder adduction and neutral rotation, with the elbow at 90° flexion, with forearm mid-rotation and supported, and also with the wrist at a neutral position. The participants were permitted to make three attempts with a 1-minute break between each attempt, and the best result was recorded as kilogram-force (kg-f).

Scapular dyskinesis frequency was evaluated statically with the lateral scapular slide test (LSST) and dynamically with the scapular dyskinesis test (SDT) [24]. Both tests were performed while the participant was standing. In LSST, the distance between the spinous process of the closest vertebra and the inferior angle of both scapulae was measured in centimeters with a tape measure. These measurements were applied in three different positions (neutral, 45°, and 90° abduction) and were considered positive if the difference between the two sides was higher than 1.5-cm in any position [25]. In the SDT, the participant holding a 500-gram dumbbell in both hands was asked to keep the arms neutral. Then, they were asked to abduct both arms 180° with the thumbs in the up position and slowly lower the arms after reaching the final degree. Dyskinesis was determined by following the scapulohumeral rhythm from the back of the participant [26].

Shoulder functionality was assessed using the American Shoulder and Elbow Surgeons (ASES) Shoulder Score. The test is presented as a 10-question ADL questionnaire completed by the patient considering their affected side, and a combined test that visually evaluates pain. A maximum of 50 points each is obtainable from the ADL and pain scores, with a maximum of 100 points in total. Participants' shoulder functionality increases in accordance to increases in the participants' scores. Each question in the ADL questionnaire is assessed according to a 4-point, Likert-type scale that is scored as 0 (*I cannot do it at all*), 1 (*I find it very hard to do*), 2 (*I only have mild difficulty*), and 3 (*It is not difficult*). The participants' level of pain was assessed using the Visual Analog Scale, which ranges from 0 (*worst pain*) to 50 (*no pain*). The physical and mental component summaries (PCS and MCS) of the 36-Item Short Form (SF-36) Health Survey were used to assess the participants' QoL. Each subdivision score is calculated by averaging the related fields and ranges from 0 (*worst QoL*) to 100 (*best QoL*).

## Statistical analysis

Categorical variables such as gender, dominant side, literacy status, smoking habits, alcohol consumption, previous diseases, and scapular dyskinesis frequency were compared using the chi-square test and expressed as number (*n*) and percentage (%). The distribution of continuous variables was evaluated using the one-sample Kolmogorov–Smirnov test. Normally distributed variables were expressed as means ± standard deviations, whereas variables not normally distributed were expressed as medians [minimum–maximum]. The *t*-test was used for normally distributed continuous variables of age, BMI, LVEF, ROM, grip strength, and SF-36 Health Survey score. Mann–Whitney *U* test was used for continuous variables that were not normally distributed, such as ASES Shoulder Score. Statistical significance value was considered as $P < .05$. All data were analyzed using IBM SPSS Statistics version 22 (IBM Corp., Armonk, NY, USA).

## Results

The participants' demographic characteristics and clinical data are summarized as shown in Table 1. A total of 60 participants—30 in the CG and 30 in the SG—were included in the

**Table 1. Demographic characteristics and clinical data of the groups.**

| Variables | Study group (n = 30) | Control group (n = 30) | *p* Value |
|---|---|---|---|
| Age, years | 62.12 ± 13.56 | 58.23 ± 7.48 | 0.21 |
| Male gender | 21 (70) | 17 (56.7) | 0.284 |
| Right dominant | 27 (90) | 28 (93.3) | 0.64 |
| Literate | 26 (86.7) | 27 (90) | 0.688 |
| Smoking | 4 (13.3) | 9 (30) | 0.136 |
| Alcohol use | 3 (10) | 6 (20) | 0.278 |
| Comorbidity(s) | | | |
| Hypertension | 18 (60) | 11 (36.7) | 0.071 |
| Diabetes mellitus | 10 (33.3) | 4 (13.3) | 0.067 |
| Hyperlipidemia | 11 (36.7) | 5 (16.7) | 0.08 |
| Coronary artery disease | 12 (40) | 4 (13.3) | ***0.02*** |
| Heart failure | 8 (26.7) | 2 (6.7) | ***0.038*** |
| Body mass index, kg/m$^2$ | 28.9 ± 3.6 | 27.2 ± 3.4 | 0.061 |
| Left ventricular ejection fraction, % | 49 ± 11 | 55 ± 8 | 0.06 |
| Time since implantation, months | 20 ± 7 | - | - |
| Generator–clavicula distance, cm | 4.7 ± 1.1 | - | - |
| Generator–acromioclavicular joint distance, cm | 7.9 ± 2.9 | - | - |

analysis. The participants' mean age was 60.17 ± 10.52 years old, and 63.3% were male. There were no differences between the groups regarding age, gender, dominant side, literacy status, or their smoking habits and alcohol consumption. Similarly, there was no difference in the incidence of hypertension, diabetes mellitus, or dyslipidemia. The frequency of coronary artery disease (40.0% vs. 13.3%, *P* = .02) and congestive heart failure (26.7% vs. 6.7%, *P* = .038) was found to be significantly higher in the SG than in the CG. Finally, there was no difference in mean BMI and LVEF values between the two groups.

There were no significant differences found between the two groups in terms of the participants' right shoulder mean flexion and abduction ROM. Conversely, left shoulder mean flexion and abduction ROM were shown to be significantly lower in the SG than in the CG (142˚ ± 16˚ vs. 148˚ ± 16˚, *P* = .016 and 121˚ ± 18˚ vs. 134˚ ± 17˚, *P* = .001, respectively; see Fig 3).

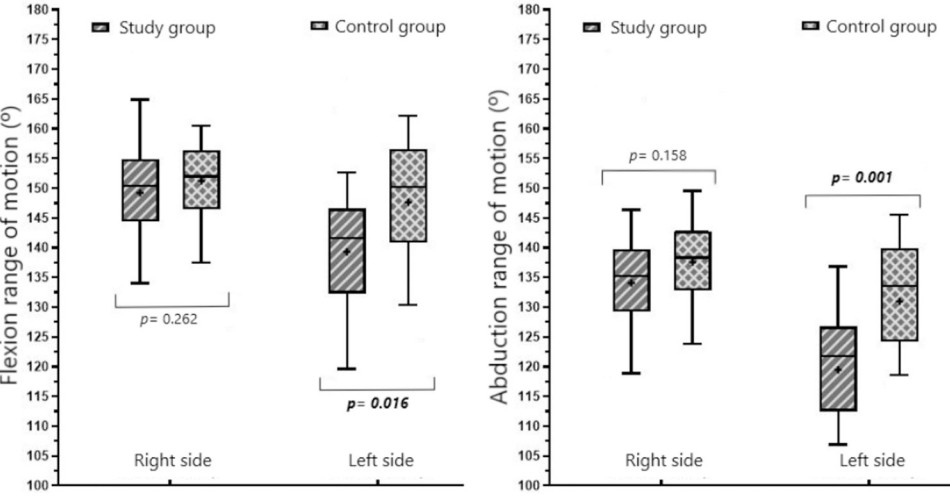

**Fig 3. Comparison of the groups' shoulder flexion and abduction range of motion values.**

**Table 2. Comparison of the groups' grip strength, scapular dyskinesis, shoulder functionality, and quality of life values.**

| Variables | Study group (n = 30) | Control group (n = 30) | P Value |
|---|---|---|---|
| Grip strength, kg-f | | | |
| Right side | 42.8 ± 11.9 | 44.6 ± 13.1 | 0.166 |
| Left side | 28.2 ± 4.6 | 33.1 ± 6.5 | *0.036* |
| Scapular dyskinesis frequency | | | |
| Lateral scapular slide test | 14 (46.7) | 3 (10) | *0.002* |
| Scapular dyskinesis test | 17 (56.7) | 4 (13.3) | < *0.001* |
| ASES Shoulder Score | 64 [52–88] | 86 [72–100] | *0.014* |
| SF-36 Health Survey | | | |
| Physical component summary | 66.8 ± 9.2 | 79.4 ± 6.4 | *0.007* |
| Mental component summary | 75.4 ± 8.2 | 76.2 ± 7.4 | 0.246 |

Abbreviations: ASES, American Shoulder and Elbow Surgeons; SF-36, 36-Item Short Form.

In comparing the participants' grip strength, a significant loss on the left side was detected in the SG when compared to the CG (28.2 ± 4.6 kg-f and 33.1 ± 6.5 kg-f, $P = .036$), but there was no difference revealed in terms of the participants' right side. Scapular dyskinesis was not detected in any of the participants in the neutral position. On the other hand, the static scapular dyskinesis frequency at abduction was shown to be significantly higher in the SG than in the CG (46.7% vs. 10.0%, $P = .002$). Moreover, the dynamic scapular dyskinesis frequency was more common in the SG than in the CG (56.7% vs. 13.3%, $P < .001$). The ASES Shoulder Score for shoulder functionality was significantly lower in the SG than in the CG (64 [52–88] vs. 86 [72–100], $P = .014$). Similarly, the PCS score from the SF-36 Health Survey was revealed to be significantly lower in the SG than in the CG (66.8 ± 9.2 vs. 79.4 ± 6.4, $P = .007$), but no difference were found in the MCS subdivision. Table 2 presents the study's results in terms of the participants' shoulder function, grip strength, scapular dyskinesis, shoulder functionality, and QoL.

## Discussion

As a result of the current study, it was determined that CIED implantation was associated with decreased ROM and grip strength on the ipsilateral shoulder, and also increased scapular dyskinesis frequency. In addition, reductions in shoulder functionality and QoL were detected.

In the literature, conditions such as decreased shoulder mobility, loss of muscular strength, increased shoulder pain and disability, and reduction in ADL and QoL have been associated with CIED implantation [11,12,19–21]. The movement limitation was mainly found in flexion and abduction, as in the current study. The most important independent determinant of shoulder impairment in CIED recipients is post-procedural pain [16]. In addition, large device size and long incision length are other predictors [27]. It is evident that the device's proximity to the pectoral muscles and shoulder joint and the avoidance behaviors of the patients to minimize the risk of device malfunction trigger shoulder disorders [19].

Disability caused by shoulder dysfunction in CIED recipients has become a significant health problem due to the increasing number of device implants. In the last decade, prospective randomized studies demonstrated that exercises to protect shoulder ROM and shoulder girdle muscular strength have been shown to be effective and safe in preventing pain and disability following CIED implantation [13–16].

As previously mentioned, the effects of CIED insertion on the ipsilateral glenohumeral joint have been clarified, but the effects on the scapulothoracic joint are unknown. Studies have shown that scapular dyskinesis can develop due to many different shoulder pathologies, including surgical sites close to the shoulder [28,29]. It is thought that muscle weakness and instability secondary to pain and immobilization may result in scapular dyskinesis [23,30]. The current study confirmed that CIED recipients exhibited significantly limited shoulder joint movements on the implant side and also reduced grip strength.

The ASES Shoulder Score is frequently used to evaluate shoulder functionality and disability in various shoulder pathologies [31,32]. In the current study, it was shown that shoulder functionality decreased following CIED implantation. In addition, decreased QoL was detected due to these types of disorders.

## Study limitations

Certain limitations to the current study are worthy of mention. Prior to CIED implantation, there was limited data available on shoulder functions. In addition, the physiotherapist in the study was not blinded to the groups, as the presence of device batteries could be easily detected through inspection. The possible adverse effects of higher frequencies of cardiovascular comorbidities in patients with CIED than in patients without CIED on the evaluation parameters cannot be ruled out in the study results. Other limitations of the study are that it included relatively few participants and was single-centered. Finally, this is a cross-sectional study and cross-sectional studies cannot establish a clear cause-and-effect relationship or analyze behavior over a period of time.

## Conclusions

Although CIEDs improve cardiac outcomes, potential complications may adversely affect patients. Among these complications, those related to the shoulder are common but are also often neglected or overlooked. This situation results in functional limitation, pain, and disability in the shoulder and reduces the QoL of patients. It would be more beneficial to consider the scapulothoracic joint together with the glenohumeral joint following CIED implantation so as to prevent these disorders from developing. This benefit may be clarified by future clinical studies involving exercises that target scapular dyskinesis. The authors believe that examining this issue through a larger sample and in more detail would yield a positive contribution to preventative shoulder rehabilitation.

## Acknowledgments

We thank all participants.

## Author Contributions

**Conceptualization:** Cansu Cosgun, Muharrem Said Cosgun.

**Data curation:** Cansu Cosgun, Muharrem Said Cosgun.

**Formal analysis:** Cansu Cosgun, Buket Buyukturan.

**Funding acquisition:** Cansu Cosgun, Muharrem Said Cosgun, Oznur Buyukturan.

**Investigation:** Cansu Cosgun, Muharrem Said Cosgun, Oznur Buyukturan, Buket Buyukturan.

**Methodology:** Cansu Cosgun, Oznur Buyukturan.

**Project administration:** Cansu Cosgun.

**Resources:** Cansu Cosgun, Muharrem Said Cosgun, Oznur Buyukturan, Buket Buyukturan.

**Software:** Cansu Cosgun, Muharrem Said Cosgun.

**Supervision:** Cansu Cosgun, Muharrem Said Cosgun, Oznur Buyukturan.

**Validation:** Cansu Cosgun, Muharrem Said Cosgun, Buket Buyukturan.

**Visualization:** Cansu Cosgun, Oznur Buyukturan.

**Writing – original draft:** Cansu Cosgun, Muharrem Said Cosgun, Buket Buyukturan.

**Writing – review & editing:** Cansu Cosgun, Muharrem Said Cosgun.

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
