## [Decision Letter · Decision Letter 0]

14 Mar 2023

PONE-D-23-01636The importance of scapular dyskinesia in cardiac implantable electronic device (CIED)-related shoulder impairmentPLOS ONE

Dear Dr. Cosgun,

Thank you for submitting your manuscript to PLOS ONE. After careful consideration, we feel that it has merit but does not fully meet PLOS ONE’s publication criteria as it currently stands. Therefore, we invite you to submit a revised version of the manuscript that addresses the points raised during the review process.

We look forward to receiving your revised manuscript.

Kind regards,

Eyüp Serhat Çalık

Academic Editor

PLOS ONE

Journal Requirements:

4. We note that Figure 2 in your submission contain copyrighted images. All PLOS content is published under the Creative Commons Attribution License (CC BY 4.0), which means that the manuscript, images, and Supporting Information files will be freely available online, and any third party is permitted to access, download, copy, distribute, and use these materials in any way, even commercially, with proper attribution. For more information, see our copyright guidelines: http://journals.plos.org/plosone/s/licenses-and-copyright.

Additional Editor Comments:

Dear Authors

I read and analyzed your manuscript with interest. Cardiac implanted devices are life-saving for many heart patients, but it seems that they have not been studied much in terms of creating movement disorders that will significantly affect the quality of life. Thank you for emphasizing this issue in your study. Your article is generally well written, but below are some suggestions from two reviewers that need to be answered. I wish you success.

Reviewers' comments:

Reviewer's Responses to Questions

**Comments to the Author**

1. Is the manuscript technically sound, and do the data support the conclusions?

Reviewer #1: Yes

Reviewer #2: Yes

2. Has the statistical analysis been performed appropriately and rigorously? 

Reviewer #1: Yes

Reviewer #2: Yes

3. Have the authors made all data underlying the findings in their manuscript fully available?

Reviewer #1: No

Reviewer #2: Yes

4. Is the manuscript presented in an intelligible fashion and written in standard English?

Reviewer #1: Yes

Reviewer #2: Yes

5. Review Comments to the Author

Reviewer #1: Thank you for the manuscript The importance of scapular dyskinesia in cardiac implantable electronic device (CIED) related shoulder impairment for review. I have read and assessed it. My thoughts are follows:

This is a specific study assessing the clinical status of the glenohumeral and scapulothoracic joints in patients with and without cardiac implantable electronic device. The design is clear. However some revisions have to be done before publication:

In the all document, the word dyskinesis should be used instead of dyskinesia.

In the all document, the word frequency should be used instead of prevalence.

In the Introduction section, the clinical importance of scapular dyskinesis should be mentioned.

In the Introduction section, a period should be used instead of a comma at the end of sentence (...procedural complications [8], Acute...).

In the Methods section, references related to determining scapular dyskinesis and evaluating statically with the lateral scapular slide test (LSST) and dynamically with the scapular dyskinesia test (SDT) should be added.

In the Results section, the word frequency should be used instead of incidence (...The incidence of coronary artery...).

In the Results section, it is seen that the frequencies of Coronary artery disease and Heart failure were higher in patients with cardiac implantable electronic device (CIED) than without. Therefore, possible negative effects of these comorbidities on the evaluation parameters cannot be ruled out in the study results. This should be mentioned in the limitations paragrahp. In addition, it should be added that this study has the advantages and disadvantages of the cross-sectional design.

Reviewer #2: This manuscript addresses the impact of an implanted cardiac device on ipsilateral shoulder range of motion, scapular mobility, pain and patient reported function. The research design, measurements, and statistical design align very well to address the research question. The abstract, background, methods, results and conclusion are well written, clear and concise. All tables are presented well, including statistical results.

Given the prevalence of cardiac implanted devices in the aging population, attention to musculoskeletal function and / or related disability is important to the individual's quality of life. This manuscript provides objective data as to the reduction in range of motion and arm strength measured on the ipsilateral side following cardiac device implantation; future studies may be able to better outline the timing and type of physical exercise to preserve range of motion and arm strength post-implantation of cardiac devices.

Recommendations:

1. Define Range of Motion measurements ( active range of motion or passive range of motion)

2. Consider change in title to "Functional impact of Cardiac implanted devices on ipsilateral shoulder range of motion, scapular mobility and self reported quality of life".

6. PLOS authors have the option to publish the peer review history of their article (what does this mean?). If published, this will include your full peer review and any attached files.

Reviewer #1: No

Reviewer #2: **Yes: **Diane W. Braza, MD

---

## [Author Response · Author response to Decision Letter 0]

17 Mar 2023

Dear Academic Editor (Eyüp Serhat Çalık),

1. We've edited our manuscript to meet PLOS ONE's styling requirements, including file naming.

2. In our Data Availability statement, we have specified where the minimum dataset can be found.

3. We've included our full ethical statement in the "Methods" section of our manuscript file.

4. We remove figure 2 from your submission.

Kind regards.

 

Dear Reviewer #1,

1. Linguistic errors in the entire document have been corrected according to your suggestions. 

2. Added references on LSST and SDT.

3. The possible negative effects of comorbidities such as CAD and CHF on the evaluation parameters are mentioned in the limitations paragrahp.

4. Disadvantages of the cross-sectional design of the study were mentioned in the limitations paragrahp.

Kind regards.

 

Dear Reviewer #2 (Diane W. Braza, MD),

1. It was defined in the ''Shoulder evaluations'' section where the range of motion measurements was ''active''.

2. The title has been changed as per your suggestion.

Kind regards.

---

## [Editor Report · Decision Letter 1]

27 Mar 2023

Functional impact of cardiac implanted devices on ipsilateral shoulder range of motion, scapular mobility, and self-reported quality of life

PONE-D-23-01636R1

Dear Dr. Coşgun,

We’re pleased to inform you that your manuscript has been judged scientifically suitable for publication and will be formally accepted for publication once it meets all outstanding technical requirements.

Kind regards,

Eyüp Serhat Çalık

Academic Editor

PLOS ONE
---

## [Editor Report · Acceptance letter]

30 Mar 2023

PONE-D-23-01636R1 

Functional impact of cardiac implanted devices on ipsilateral shoulder range of motion, scapular mobility, and self-reported quality of life 

Dear Dr. Cosgun:

I'm pleased to inform you that your manuscript has been deemed suitable for publication in PLOS ONE. Congratulations! Your manuscript is now with our production department. 

Kind regards, 

on behalf of

Dr. Eyüp Serhat Çalık 

Academic Editor

PLOS ONE